# Enhancing Wound Recovery: A Self-Gelling Powder for Improved Hemostasis and Healing

**DOI:** 10.3390/polym16131795

**Published:** 2024-06-25

**Authors:** Yuzhou Zhao, Yanni Gao, Zihao Shen, Mingze Ni, Juan Xu, Ting Wang

**Affiliations:** 1Aulin College, Northeast Forestry University, Harbin 150000, China; 1564610285@nefu.edu.cn (Y.Z.); 2021224742@nefu.edu.cn (Y.G.); zihaoshenhugh@163.com (Z.S.); 1125785848@nefu.edu.cn (M.N.); 2College of Chemistry, Chemical Engineering and Resource Utilization, Northeast Forestry University, 26 Hexing Road, Harbin 150040, China; 3NHC Key Laboratory of Reproductive Health Engineering Technology Research, Haidian District, No. 12, Da Hui Si Road, Beijing 100081, China; 4National Research Institute for Family Planning, Haidian District, No. 12, Da Hui Si Road, Beijing 100081, China

**Keywords:** carboxymethylcellulose calcium, lysine, hemostasis, healing

## Abstract

A novel self-gelatinizing powder was designed to accelerate wound healing through enhanced hemostasis and tissue recovery. Significantly, this research addresses the critical need for innovative wound management solutions by presenting a novel approach. Carboxymethylcellulose calcium (CMC-Ca) was synthesized using an ion exchange method, and lysine (Lys) was integrated through physical mixing to augment the material’s functional characteristics. The prepared powder underwent comprehensive evaluation for its self-gelling capacity, gelation time, adhesion, swelling rate, coagulation efficiency, hemostatic effectiveness, and wound healing promotion. Results indicate that the self-gelatinizing powder exhibited remarkable water absorption capabilities, absorbing liquid up to 30 times its weight and achieving rapid coagulation within 3 min. The inclusion of Lys notably enhanced the powder’s gel-forming properties. The gelation time was determined to be within 4 s using a rotational rheometer, with the powder rapidly forming a stable gel on the skin surface. Furthermore, in a mouse skin injury model, near-complete skin recovery was observed within 14 days, underscoring the powder’s impressive self-healing attributes and promising application prospects in wound management.

## 1. Introduction

Hemorrhage is a leading cause of trauma-related deaths, accounting for approximately 30–40% of all such fatalities [1]. This underscores the critical need for efficient hemostatic materials in emergency medical care settings. The ideal hemostatic materials should meet several essential criteria: rapid absorption of blood moisture to foster the aggregation of blood cells and coagulation factors; activation of the body’s coagulation cascade; and strong adherence to the wound site to establish an effective physical barrier.

Currently, commonly utilized hemostatic materials include gauze [2], sponge [3], hydrogel [4], and powder [5]. Mechanical compression with gauze and sponges is a traditional yet effective method for stemming bleeding by promoting blood cell coagulation [6,7]. However, their lack of adhesive properties necessitates continuous pressure application, potentially compromising wound integrity and reducing hemostatic efficacy.

Hydrogels represent a significant advancement in hemostatic technology, offering superior adhesion compared to traditional materials [8]. Nevertheless, their effectiveness can be compromised in the presence of blood, which can reduce their adhesion to tissues and limit their use in emergency scenes [9].

Powders have been widely used as hemostatic materials due to their high surface area, which facilitates direct contact between the hemostatic components and the body, unlike other forms of hemostatic materials [10,11,12]. Conventional hemostatic powders, such as WoundStat^TM^ [13,14] (montmorillonite-based), have demonstrated significant hemostatic effects. However, they also present notable drawbacks. For instance, zeolite-based powders can clog their surface pores upon contact with blood, diminishing their coagulation effectiveness. Montmorillonite-based materials pose risks of entering blood vessels and potentially causing serious complications like blood clots, leading to FDA restrictions on their use.

To overcome these limitations, researchers have explored self-gelling powders that absorb blood to form in situ hydrogels, providing both effective hemostasis and a physical barrier at the wound site due to their excellent adhesive properties [15]. For example, a self-gelling hemostatic powder developed by Peng et al. [9] consists of polyethyleneimine, poly (acrylic acid), and quaternized chitosan (PEI/PAA/QCS). It possesses rapid water absorption capabilities that enable it to form a gel, along with excellent adhesion properties. However, the inclusion of polyethyleneimine raises concerns regarding potential biological toxicity. The quest for hemostatic powders that are both effective and biocompatible continues to be a crucial research area.

Carboxymethyl cellulose sodium (CMC-Na) is a natural cellulose derivative with good biocompatibility [15], great water solubility [16,17], and degradability [18], which can form hydrogels by physical cross-linking [19,20]. While CMC-Na serves as an effective blood absorbent, it lacks the clot-promoting function of calcium ions. Therefore, by incorporating calcium ions into CMC to form CMC-Ca, its coagulation-promoting efficacy can be enhanced, thus better fulfilling the purpose of hemostasis. However, previous studies on CMC-Ca-prepared powders had poor dissolution properties and did not address wound healing after skin hemostasis [21]. Integration of Lys, known for its wound healing properties [22,23], remains an underexplored area.

Building on prior research on CMC [24], this study introduced Ca^2+^ to enhance its coagulation capabilities, resulting in CMC-Ca. By combining CMC-Ca with Lys, CMC-Ca-Lys was formulated as a powder capable of direct interaction with blood, rapidly absorbing it, and forming an in situ hydrogel. This innovative powder offers strong adhesive properties, creating a physical barrier while supporting tissue recovery. Our research aims to provide a novel approach to hemostatic material development, effectively promoting both wound hemostasis and healing.

## 2. Materials and Methods

### 2.1. Materials

Carboxymethyl cellulose sodium (CMC-Na, MW 700,000, degree of substitution (DS) = 0.9) was purchased from Macklin Biochemical Co., Ltd. (Shanghai, China) Anhydrous CaCl_2_ was provided by Tianda Chemical Reagent Factory. Lysine (Lys, AR) was purchased from Dibo Biotechnology Co., Ltd. (Shanghai, China) Silver nitrate (AgNO_3_, AR) was purchased from Guangfu Technology Development Co., Ltd. (Tianjin, China) Ethanol and phosphate-buffered saline were obtained from Rhawn Co., Ltd. (Shanghai, China).

### 2.2. Preparation and Characterization of CMC-Ca-Lys Powder

To prepare CMC-Ca powder, 5 g of sodium carboxymethylcellulose powder was mixed with 200 mL of ethanol/water solution (75 wt%) containing 15 wt% CaCl_2_. This mixture was stirred at room temperature for 3.5 h, facilitating the ion exchange process to introduce Ca^2+^ into the CMC-Na structure. Following the reaction, the mixture was centrifuged at 3500 rpm, and the precipitate was washed with ethanol three times to remove unreacted CaCl_2_. The clarity of the supernatant was then tested using AgNO_3_ to confirm the absence of Cl^−^. Once verified, the precipitate was oven-dried at 40 °C for 24 h, yielding CMC-Ca powder.

Subsequently, the CMC-Ca powder was mixed with various ratios of Lys to obtain the final product, CMC-Ca-Lys. The resultant mixture underwent grinding and sieving through a 200-mesh sieve to ensure a homogeneous, fine powder suitable for wound application.

The Ca^2+^ content in CMC-Ca-Lys powder was determined using atomic absorption spectroscopy. The powder was dissolved in deionized water, and its calcium ion content, along with sodium ion content, was measured using an atomic absorption spectrometer (TAS-986, Puxi General Instrument Co., Ltd., Beijing, China). The scanning electron microscope JSM-7500F (JEOL Co., Ltd., Tokyo, Japan) was used for the observation of the microstructure of the powder, and the composition of the powder was characterized by energy-dispersive X-ray spectroscopy (EDS).

### 2.3. Assessing the Swelling and Self-Healing Capacities of In Situ Hydrogels

To investigate the swelling properties of the hemostatic powder, various compositions of CMC-Ca-Lys powder were prepared, each containing different Lys contents: 50.00%, 20.00%, 16.66%, 12.50%, 10.00%, and 8.333%. Each variant, weighing 0.1 g, was mixed with 2.00 mL of 0.01 mol L^−1^ PBS solution to form an in situ hydrogel. Subsequently, these hydrogels were immersed in 3.00 mL of PBS solution and allowed to stand at room temperature. At regular intervals of 0.5 h, the hydrogels were removed to measure their swollen weight. Afterward, the samples were dried to determine their dry mass. The obtained data were used to construct the swelling kinetics curve. The formula for calculating the swelling ratio is represented by Equation (1).
(1)$$P=\frac{m_{1}-m_{0}}{m_{0}}\times 100\%$$
(2)$$D=\frac{m_{1}-m_{n}}{m_{n}}$$
where *m*_1_ is the mass of the hydrogel in its swelling state, *m*_0_ is the mass of the hydrogel after drying, and *m_n_* is the mass of the pre-prepared powder.

The self-healing capability of hydrogels plays a pivotal role in their effectiveness for hemostatic applications, enabling the hydrogel to preserve its integrity and functionality under various external conditions. To evaluate the hydrogel’s self-healing capacity, a sample of the hydrogel was precisely divided into two halves using a razor blade. After separating, the two halves were carefully realigned and reconnected. The subsequent reintegration and healing process of the hydrogel were then monitored, with evaluations conducted after a thirty-minute period to assess the extent of self-healing.

### 2.4. Characterizing Rheological and Adhesion Properties

In rheological testing, samples underwent analysis using plates with an 8.0 mm diameter and 1.0 mm spacing. Temporal strain scans were performed using 1% strain in the frequency range of 0.1–50 Hz by a rotational rheometer (AR-G2 TA Instruments, New Castle, DE, USA). Meanwhile, powders were placed on parallel plates at 37 °C with a 0.1 mol L^−1^ PBS solution added. Time-scan tests were then carried out at a constant frequency of 1 Hz and 1% strain to assess the energy storage modulus (G′) and loss modulus (G″) in order to determine the gelation time.

Adhesion properties were evaluated by applying powder-formed hydrogels to fresh pigskin. The pigskin underwent thorough washing with deionized water and ethanol before being utilized. Rhodamine B-stained hydrogels were then placed onto the prepared skins, and adhesion properties were observed through skin inversion and twisting maneuvers.

### 2.5. In Vitro Coagulation Test

Based on a previous study [25], the coagulation ability of the powder was assessed using the coagulation index reaction. Specifically, 10 mg of CMC-Ca-Lys powder was evenly dispersed at the bottom of the test tube. Subsequently, a mixture of 180 μL of anticoagulated pig blood and 20 μL of 0.2 mol L^−1^ CaCl_2_ was swiftly added to the test tubes. The control groups consisted of blank experimental tubes without any anticoagulant. The test tubes were shaken at 37 °C. At specified time intervals (0.5 min, 1.0 min, 2.0 min, and 3.0 min), the test tubes were removed, and 10.0 mL of deionized water was slowly added to separate uncoagulated erythrocytes. The resulting supernatant was separated, and the absorbance of hemoglobin at 540 nm was measured by UV–Vis spectrophotometer (TU1901, Beijing Puxi General Instrument Co., Ltd.). Subsequently, the coagulation index was calculated according to Equation (2).
(3)$$\mathrm{BCI}\;(\mathrm{Blood}\;\mathrm{clotting}\;\mathrm{index})\;=\frac{A_{s}}{A_{0}}\times 100\%$$
where $A_{s}$ is the absorbance of the supernatant of the sample group and $A_{0}$ is the absorbance of the supernatant of the control group.

### 2.6. In Vivo Hemostasis Test

All animal experiments were conducted in accordance with the UK Animals (Scientific Procedures) Act 1986 and related guidelines, the European Union Directive on Animal Experimentation 2010/63/EU, or the National Research Council of the Northeast Forestry University Guidelines for the Care and Use of Laboratory Animals (Approval No. 2022082609). A mouse dorsal skin trauma model was employed for this study, with the protocol intentionally excluding the involvement of the mice’s sexual organs. This measure was taken to minimize potential variations between individual mice and their sexes that could introduce experimental errors.

The in vivo hemostatic ability of the CMC-Ca-Lys powder was tested by a mouse tail amputation experiment. Several female Kunming mice (20 ± 1 g) of similar weight and morphology were taken, and their tails were placed on filter paper and then excised at the same location (about 2/3 of the tail). After excision, the sample group was hemostatized using CMC-Ca-Lys powder and gauze, while the control group was left untreated. The amount of blood loss was determined by testing the weight of the filter paper.

### 2.7. In Vivo Wound Healing Test

Nine female Kunming mice of the same morphology were used for this study. After feeding the mice individually for 14 days, a wound of 12 mm in diameter on the back of the mice was made. The mice were divided into three groups. The sample group was treated with a hydrogel formed by CMC-Ca-Lys powder (Lys group) and CMC-Ca powder (CMC group), and the control group was treated with medical gauze. The wounds were observed and photographed at predetermined times (day 1, day 5, day 7, and day 14) to record their recovery status. The staged wound size was calculated to derive the wound recovery rate on day 1, day 5, day 7, and day 14, and the wound healing rate was compared between the different conditions. The histological changes in the skin of tissue sections near the wounds of mice (after 14 days of recovery) were analyzed by hematoxylin-eosin (H&E) staining and Masson staining. Three mice using the Lys group were executed on the 14th day, and their hearts, livers, spleens, lungs, and kidneys were sectioned for observation.

### 2.8. Statistical Analysis

Statistical methods were used to compare group differences and assess their statistical significance. Utilize the mean standard deviation for the analysis of data collected from distinct parallel surveys. Perform statistical analysis using Office 2023 and Origin software 8.5, with a sample size of *n* = 3. The significance level is set at * *p* < 0.05, indicating statistical significance; ** *p* < 0.01 denotes high statistical significance; *** *p* < 0.001 signifies the highest level of statistical significance.

## 3. Result and Discussion

### 3.1. CMC-Ca-Lys Powder Analysis

The synthesis of CMC-Ca-Lys powder involved a two-step process. Initially, calcium-substituted calcium carboxymethyl cellulose powder was prepared through ion exchange, resulting in a calcium content of 6.21 mg g^−1^, as confirmed by atomic absorption spectroscopy (Figure 1A). Subsequently, Lys was incorporated into the calcium carboxymethyl cellulose matrix, leading to the formation of CMC-Ca-Lys powder.

Scanning electron microscopy was employed to analyze the structure of the synthesized CMC-Ca-Lys powder. The images revealed a rod-like structure for the CMC-Ca-Lys powder (Figure 1B), while Lys exhibited a granular morphology. Notably, SEM analysis confirmed the homogeneous distribution of both components within the synthesized powder. 

FTIR spectra showed that after ion exchange, the resulting CMC-Ca, CMC-Na, and CMC-Ca-Lys displayed essentially identical spectral profiles. This proves that the chain structure is essentially unchanged, but the -COO^−^ group at 1407 cm^−1^ to 1425 cm^−1^ shows a symmetric stretching vibration due to the coordination of calcium ions. The CMC-Ca-Lys, with the addition of Lys, has an NH_2_ absorption peak at 3110 cm^−1^, proving the successful incorporation of Lys. Additionally, Figure 1D–F present the Ca and Na content, respectively, further validating the successful synthesis of CMC-Ca-Lys powder. Figure 1D shows that the elemental content of calcium is much higher than that of sodium, and the initial CMC powder is free of sodium ions, which is a good indication of the successful exchange of calcium and sodium ions. These compositional analyses support the structural integrity and homogeneity of the synthesized powder.

Together, these findings underscore the potential of CMC-Ca-Lys powder for applications in wound management and hemostasis, highlighting its structural integrity and efficacy.

### 3.2. Swelling Properties

Swelling properties represent a critical aspect of hydrogel functionality, particularly in wound management. A hydrogel’s capacity to swiftly absorb exudate from wounds and water from the blood directly impacts its efficacy in hemostasis and wound healing promotion.

The swelling properties and hemostatic applications of CMC-Ca have been extensively studied [26], yet research on incorporating Lys into the aforementioned system is lacking. Notably, our investigation reveals that the addition of Lys significantly influences the swelling performance of the hydrogel (Figure 2A), as depicted in the dynamic swelling kinetics experiment results shown in Figure 2B.

As shown in Figure 2B, all hydrogels exhibited rapid absorption and dissolution within 0.5 h, followed by a slower but continued water absorption rate over the subsequent 2 h. Notably, hydrogels containing 50% Lys by mass displayed higher water absorption compared to their counterparts with a lower Lys content. Specifically, hydrogels with 50% Lys reached maximum swelling at approximately 2.5 h, measuring 22.56 ± 0.26. Conversely, hydrogels with reduced Lys content demonstrated enhanced swelling performance, with the highest swelling observed at 34.93 ± 0.26 for hydrogels containing 10% Lys. However, beyond a certain threshold, a reduction in Lys content led to diminished swelling performance, as evidenced by hydrogels with 8.33% Lys displaying a swelling ratio of (32.09 ± 0.19) lower than that of hydrogels with 10.00% Lys. When the Lys content was reduced to 0%, the swelling rate of the gel was not much different from that at 8.33% content.

These findings highlight the significant impact of Lys addition on the swelling behavior of the hydrogel. The optimal range of powder Lys content was determined to be between 12.50% and 8.33%, where Lys augmentation enhances the hydrogel’s hydrophilic properties, thereby augmenting its water absorption capacity and swelling performance. Consequently, the CMC-Ca-Lys powder featuring 10% Lys content was selected for subsequent experiments due to its exemplary swelling properties and ease of preparation through proportional mixing during production. Hydrogels exhibiting a swelling rate within this range hold promise as effective wound aids, proficiently absorbing exudate from the skin to facilitate optimal wound healing [27].

### 3.3. Self-Gelation Capability and Degradation Text

As shown in Figure 2C, when 50 mg of powder was sprinkled on 500 μL of PBS solution, all of the Lys powder dissolved, while CMC-Ca formed a hydrogel with CMC-Ca-Lys powder. CMC-Ca-Lys exhibited superior water absorption, attributed to the inclusion of Lys, resulting in the formation of a homogeneous transparent gel for all powders. In contrast, the gel formed by CMC-Ca showed a small portion of powder that did not transition into gel form. This observation may be attributed to the rapid water absorption of CMC-Ca, leading to the formation of a gel that blocked the contact of the internal powder with water. As shown in Figure 2D, the gel immersed in saline dissolved rapidly in the first 2 h, began to show degradation at the 4th hour, and degraded by 13.73%. Between 4 and 8 h, the gel showed rapid degradation, reaching 96.87% by the 8th hour. This experiment demonstrates that the powder can degrade on its own after forming the gel, thus avoiding secondary damage to the wound that may be caused by removing the gel.

### 3.4. Rheology, Adhesion and Self-Healing Properties 

The rheological characteristics of the hydrogel formed by the hemostatic powder were analyzed to assess its viscoelastic behavior. As depicted in Figure 3A, the gelation time of the powder after absorption of PBS solution was measured by a rheometer at 37 °C. Notably, the storage modulus (G′) curve intersected the loss modulus (G″) approximately 5 s after the addition of PBS solution to CMC-Ca-Lys, indicating rapid gelation within 4 s. Furthermore, Figure 3B displays the storage modulus (G′) and loss modulus (G″) as a function of frequency for 0.1 mg of CMC-Ca-Lys in absorbing 35 times the mass of PBS to form a gel. The consistent dominance of the storage modulus (G’) over the loss modulus (G″) signifies the formation of a hydrogel with a stable elastic state.

Adhesion properties were evaluated by examining the attachment of the rhodamine B-stained hydrogel to pig skin, as shown in Figure 3C. The hydrogel demonstrated excellent adhesion properties, firmly adhering to the skin surface.

Moreover, Figure 3D, not explicitly mentioned in the initial discussion, demonstrates the evaluation of the hydrogel’s self-healing performance through cutting experiments, wherein the two pieces of hydrogel re-joined seamlessly, validating its robust self-healing capability.

Together, these observations highlight the powder-formed hydrogel’s ability to maintain stability on the skin, showcasing excellent adhesive properties. Moreover, its remarkable self-healing ability suggests suitability for use in diverse external environments [28,29].

### 3.5. In Vitro Clotting Analysis

In vitro clotting analysis is crucial in evaluating the efficacy of hemostatic materials, as rapid hemostasis is essential for facilitating wound healing, particularly during the initial phase of hemostasis [30]. The coagulation performance of the hemostatic powder was assessed using the coagulation index (BCI) [25], with results depicted in Figure 4. The control group served as a blank experimental group for comparison.

As illustrated in Figure 4, a significant reduction in supernatant concentration was observed just 1 min after the addition of the hemostatic powder, with complete coagulation achieved within 3 min (where absorbance approached 0). In contrast, the control group’s supernatant maintained a high hemoglobin absorbance level. Several factors contribute to the coagulation effectiveness of the powder, including:(i)The hemostatic powder’s ability to absorb blood water and form an in situ gel promotes the aggregation of coagulation factors.(ii)The presence of Ca^2+^ in the hemostatic powder facilitates the formation of a coagulation factor–calcium–phospholipid complex, acting as a bridge on phospholipid surfaces of coagulation factors and platelets [30,31].(iii)The dense voids within the in situ hydrogel formed by the hemostatic powder effectively capture erythrocytes and platelets, promoting their aggregation and facilitating blood coagulation [32,33].

These findings underscore the powder’s significant potential for promoting rapid and effective hemostasis, which is pivotal for initiating and accelerating the wound healing process.

### 3.6. In Vivo Hemostasis Analysis

In vivo hemostasis analysis involved testing the powder’s ability to staunch bleeding using a mouse tail-breaking model, with experimental outcomes depicted in Figure 5A. The results revealed that hemostasis was achieved within 5 s in mice treated with CMC-Ca-Lys hemostatic powder. By the 5 min mark, hemostasis was complete, with minimal blood loss observed, notably less than that in the gauze hemostasis and blank experimental groups.

This remarkable hemostatic efficacy can be attributed to the rapid self-gelling ability and exceptional coagulation properties of CMC-Ca-Lys hemostatic powder. The self-gelling capability enables swift blood absorption, while the hydrogel’s superior viscoelasticity and tissue adhesion ensure secure fixation on the skin surface, effectively covering the wound and minimizing blood loss.

From Figure 5B, the amount of blood loss in mice using different hemostatic methods can be analyzed. Mice treated with CMC-Ca-Lys lost a total of 12.1 ± 0.1 mg of blood, whereas those subjected to gauze hemostasis experienced a total blood loss of 59.6 ± 0.2 mg, and the blank experimental group lost 91.7 ± 0.2 mg of blood. These results unequivocally demonstrate the outstanding in vitro hemostatic efficacy of CMC-Ca-Lys, showcasing its ability to effectively address bleeding in vitro.

### 3.7. In Vivo Evaluation of Wound Healing Properties

To illustrate the wound healing-promoting abilities of hydrogels formed by hemostatic powders, experiments were conducted using a mouse skin injury model. The hydrogel formed by the powder containing Lys, with optimal solubility properties, was selected for in vivo coagulation property testing. Mice were photographed on days 1, 5, 9, and 14, and their wound recovery progress is depicted in Figure 6A.

The wound healing rate, as observed in the figure, demonstrated significant progress in the Lys group compared to the control group post the 9th day. Specifically, the wound healing rates for the Lys and CMC groups were 38.9% and 81.6%, respectively, while the blank control group exhibited a rate of only 25.6%. By the 14th day, wounds in the Lys group had substantially healed, with emerging newborn hair, contrasting with the slower recovery observed in other groups. These findings underscore the remarkable wound healing potential of Lys-infused hemostatic powder, highlighting the importance of Lys supplementation in wound healing formulations.

Furthermore, tissue sections from the heart, liver, spleen, lungs, and kidneys of mice in the Lys group were examined (Figure 6F), revealing intact cells and no tissue damage. This indicates the safe application of Lys for wound treatment without adverse effects on other bodily tissues and organs.

To comprehensively evaluate the wound healing-promoting abilities of hydrogels, Masson staining and H&E staining analyses were conducted on newly formed skin tissue of mice on day 14. The specific results are presented in Figure 6D,E, observed under high and low magnification, respectively.

As demonstrated in Figure 6D, skin sections of mice from the three groups exhibited significant differences in granulation tissue thickness, with the control group showing the highest fibroblast count. Conversely, mice treated with CMC-Ca and CMC-Ca-Lys displayed denser collagen fiber presence, indicative of enhanced wound healing. In Figure 6E, the control group exhibited the highest inflammatory cell count and the slowest wound recovery, while CMC-Ca-Lys displayed the lowest inflammatory cell count and the most improved wound recovery. Powder 2 exhibited moderate wound recovery with fewer inflammatory cells. These results underscore the profound wound healing-promoting effects of Lys supplementation, attributed to enhanced collagen production and angiogenesis promotion.

Additionally, tissue section staining of major metabolic organs treated with Powder 1 revealed no significant organ damage or abnormal cell lesions, affirming the excellent biocompatibility and promising application potential of the powder.

## 4. Conclusions

In summary, an efficient hemostatic powder has been developed through a simple preparation method. The powder demonstrates rapid blood absorption and coagulation within a short time, along with a favorable gelation time as determined by rheological analysis. Importantly, all components of this hemostatic powder are non-toxic, biocompatible, and biodegradable in the human body.

Moreover, the hydrogel formed by the powder exhibits strong adhesion and remains on the skin for an extended period, providing additional moisturizing effects. The efficacy of the powder in promoting hemostasis was confirmed through a mouse tail-breaking experiment, demonstrating its ability to promptly form a gel in wounds and facilitate blood coagulation to stop bleeding.

Additionally, results from a mouse cortex injury model highlight the powder’s potential in wound healing, particularly attributed to the presence of Lys, which effectively promotes this process. Overall, these findings suggest that the hemostatic powder offers a promising avenue for the development of self-gelling systems, with significant implications for wound hemostasis and healing.

## Figures and Tables

**Figure 1 polymers-16-01795-f001:**
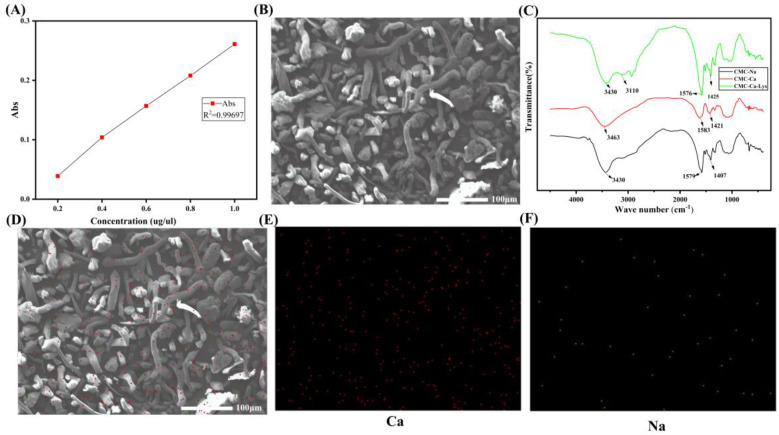
Powder characterization (**A**) standard curves for flame atomic absorption spectroscopy (R^2^ = 0.998). (**B**) SEM images of CMC-Ca-Lys powder. (**C**) FTIR spectra of CMC-Na, CMC-Ca, and CMC-Ca-Lys. (**D**) Distribution of calcium and sodium ions on powders. (**E**) Calcium content. (**F**) Sodium content.

**Figure 2 polymers-16-01795-f002:**
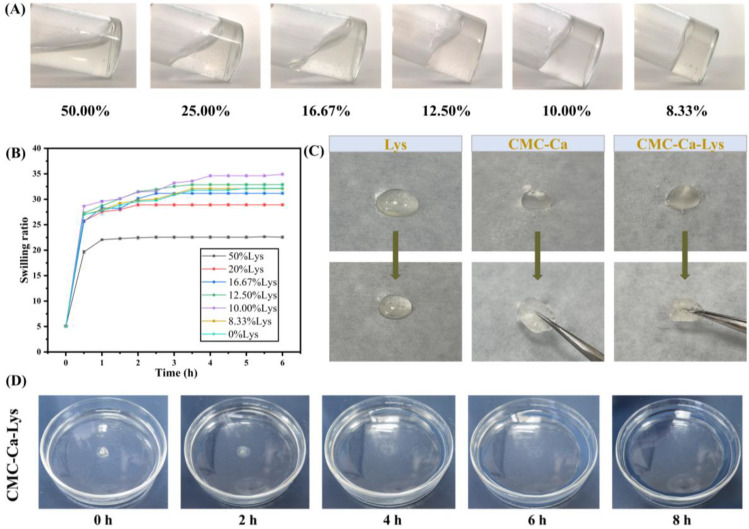
Swelling properties of hydrogel. (**A**) State diagram of gel formation by adding 3.5 mL of PBS to powders with different Lys mass ratios. (**B**) Solvation kinetics curves of hydrogels formed from powders with different Lys mass ratios. (**C**) Illustration of Lys powder, CMC-Ca powder, and CMC-Ca-Lys powder with water. (**D**) The degradation tests of CMC-Ca-Lys.

**Figure 3 polymers-16-01795-f003:**
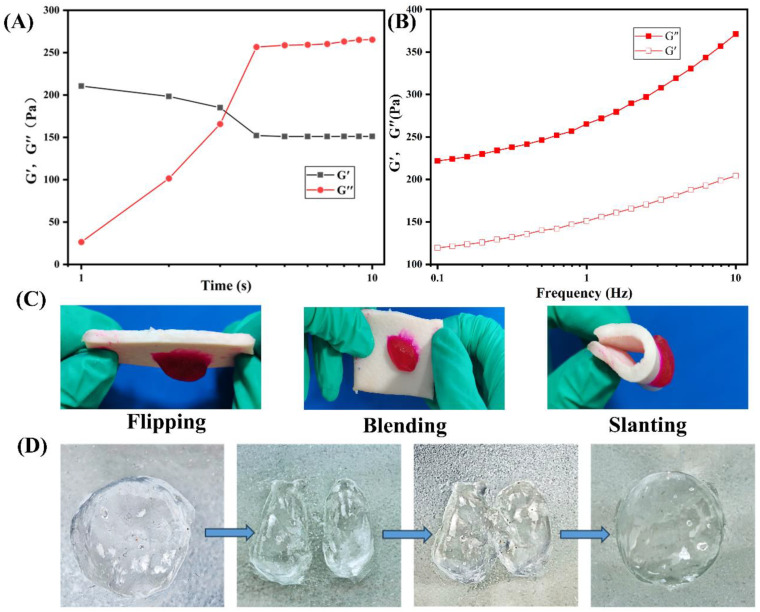
Gel characteristics (**A**) Schematic diagram of powder gel formation on porcine skin. (**B**) Schematic illustration of gel adhesion on porcine skin. (**C**) The gelation time of the powder after absorption of the PBS solution was measured by a rheometer at 37 °C. (**D**) Storage modulus (G′) and loss modulus (G″) as a function of frequency for 0.1 mg of CMC-Ca-Lys in absorbing 35 times the mass of PBS to form a gel.

**Figure 4 polymers-16-01795-f004:**
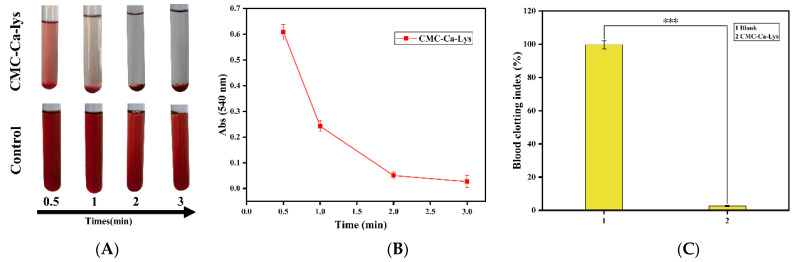
Testing of the hemostatic properties of powders. (**A**) Photographs of whole blood coagulation at different times in CMC-Ca-Lys and blank groups. (**B**) Kinetic profile of CMC-Ca-Lys whole blood coagulation in vitro. (**C**) Blood clotting index of full blood with CMC-Ca-Lys powder; *** *p* < 0.001.

**Figure 5 polymers-16-01795-f005:**
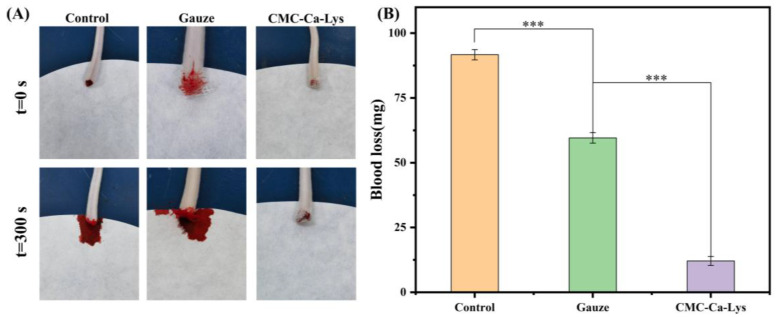
In vivo hemostatic test of powders. (**A**) Pictures of hemostasis experiments in mice. (**B**) Blood loss data in mice. Error bar indicates SD (n = 3), *** *p* < 0.001.

**Figure 6 polymers-16-01795-f006:**
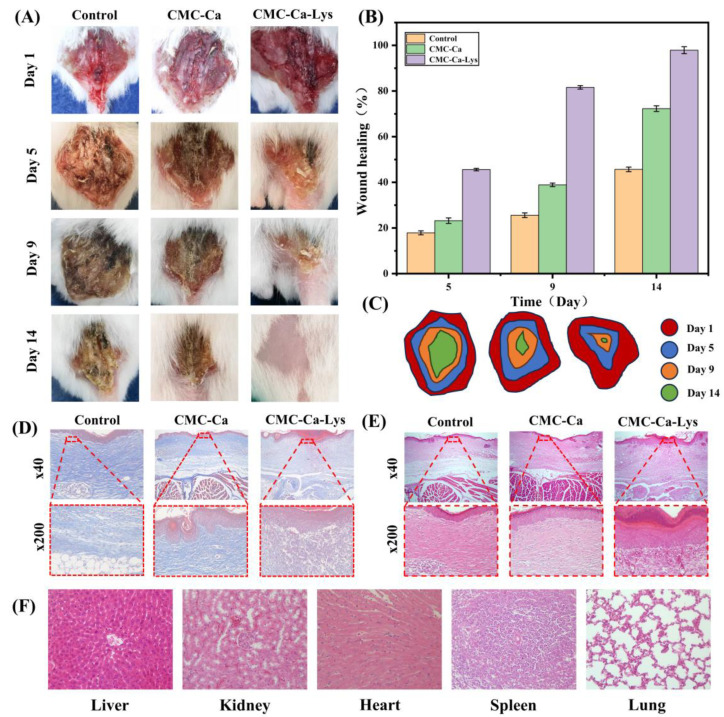
In vivo evaluation of wound healing. (**A**) Wound appearance pictures of the control group, CMC-Ca group, and CMC-Ca-Lys group. (**B**) Wound healing data. (**C**) Wound healing model. (**D**) Matson stained the image of wound tissue on day 14. (**E**) H&E stained images of wound tissue on day 14. (**F**) H&E-stained tissue sections of major organs (heart, liver, spleen, lung, and kidney).

## Data Availability

The original contributions presented in the study are included in the article, further inquiries can be directed to the corresponding author.

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
