# Peer review of "Enhancing Wound Recovery: A Self-Gelling Powder for Improved Hemostasis and Healing"

_polymers, 2024, doi:10.3390/polym16131795_

Round 1

Reviewer 1 Report

Comments and Suggestions for Authors

The paper “Enhancing Wound Recovery: A Self-Gelling Powder for Improved Hemostasis and Healing” describes the use of a powder that gelates upon contact with blood reducing blood loss and increase healing.

The subject is certainly interesting but the paper has some flaws that need attention.

Some introduction on what Lys has on the coagulation cascade may help a lot.

Starting with fig 1, the scanning EM pictures are hardly relevant because they do not show the final product. The EDX pictures are hard to interpret at the moment, much better would be a Ca/N ratio before and after the reaction. Fig 1e is a complete artefact from the sample preparation method. Upon freezing water it will form ice crystals which will push all material including the gel fibers into the regions where the growing ice crystals meet and give a pattern of boundaries of ice crystals.

The swelling as displayed in fig 2 does not seem to follow a certain trend but is random for different Lys concentrations. Is in all samples the amount of CMC-Ca equal?? Furthermore, I miss the 0% Lys in the graph.

I can hardly belief that the control in the in vitro clotting analysis does not coagulate at all, or does this sample lack Ca? a proper control will have Ca added as well. And why is this test not performed with different amounts of Lys as in the previous experiment.

The pictures of fig 5 seem to be mixed up because I see no blood loss in the control group. In the material and method, I cannot find any ethical guidelines regarding this somewhat cruel experiment from which I wander if this allowed within the EU.

Reviewer 2 Report

Comments and Suggestions for Authors

The manuscript describes a self-gelatinizing powder, obtained combining CMC-Ca with Lys, capable of direct interaction with blood, rapidly absorbing it and forming an in situ hydrogel.

The manuscript is well written and results are clearly discussed. Some major points have to be considered to improve the quality of the manuscript.

1-      In the Introduction section, please add the following recent reference:

Yao, Kaitao et al. “Superwettable calcium ion exchanged carboxymethyl cellulose powder with self-gelation, tissue adhesion and bioabsorption for effective hemorrhage control” Chemical Engineering Journal, Volume 4811 (2024) Article number 148770

The authors are invited to compare the results reported in this study with those relevant to their material.

2-      Scanning electron microscope was employed to characterize by EDS the developed material. This technique is very poor in chemical characterization. Authors assessed that the presence of “Ca and Na content, respectively, further validating the successful synthesis of CMC-Ca-Lys powder.”: this is not true. The successful of the synthesis can be verified by other analytical techniques like SSNMR, XPS, FTIR that provide information on the chemical structure of the material. The authors are kindly invited to characterize their material using one or more of these techniques.

3-      Degradation tests in opportune media, simulating wound microenvironment, have to be performed to gain information on the materials degradability.

Comments on the Quality of English Language

English language is clear

Round 2

Reviewer 1 Report

Comments and Suggestions for Authors

The paper “Enhancing Wound Recovery: A Self-Gelling Powder for Improved Hemostasis and Healing”

Improved but is certainly not ready for publication in my opinion.

I keep having trouble with figure 1. A calibration curve as displayed in fig 1a is not a relevant finding and can be presented in the SI. Also the structure of the powder after modification is hardly relevant to the properties of the gel when swollen or to the hemostatic effect. The FTIR spectra makes sense because this showed that the ion exchange worked out as it should and is therefore the first real result. For the EDX result in fig 1d,e I fail to see anything….. only after increasing the brightness of the picture I was able to see red and green dots, but I keep failing to see how this is related to the SEM picture in fig 1b even after manually overlaying the two pictures there seem no correlation.  The only way this is helpful for the reader is a real overlay of the elemental maps on top of the SEM picture, which then can be combined in one picture. With respect to the SEM picture of the gel (1f) there can be no other conclusion then that this is an artefact of the method. I am really surprised that all SEM pictures using this method are the same irrespective of the material that is used, a porous structure. Upon freezing water, it will crystalize leaving all the other material in the boundaries between the crystals, leaving an almost hexagonal structure behind. There are only two methods to avoid water from crystallization, freezing very thin films in liquid ethane or propane or high pressure freezing. The first method was rewarded the Nobel prize to Jacque Dubochet and only works for very this films, less than 1 micron. The second works for larger objects and requires freezing at 2000 bar pressure, but if the material is later warmed to -100 for sublimation the water will crystalize again with the same devastating effect. I am well aware that the method is used frequently if not throughout the entire field but that does not make it a useful method unfortunately it only shows ignorance to physics.

My suggestion is to stay away from the characterization of the gel in water and focus on the hemostatic effects which is more than interesting enough.

It is a pity that the 0%lysine results are not included especially because they are of scientific interest. In a scientific paper like this one, the product requirements should be of minor importance.

I am happy to see the corrected fig 5

Reviewer 2 Report

Comments and Suggestions for Authors

Dear authors, probably there was a misunderstanding on my request on degradation tests. 

"Degradation tests in opportune media, simulating wound microenvironment, have to be performed to gain information on the materials degradability."

I never asked for degradation tests in vivo but in vitro. You said you had done them but I don't see the results of these tests in the manuscript. I am wrong?

Author Response

请参阅附件。

Round 3

Reviewer 1 Report

Comments and Suggestions for Authors

I am very happy with the made changes in the document. Only after changing the brightness of figure 1 I was able to see the dots in fig 1d-f. please adjust before printing.